

# Local thermal gradient and large-scale circulation impacts on turbine-height wind speed forecasting over the Columbia Basin

Ye Liu[1], Yun Qian[1], and Larry K. Berg[1]

[1]Pacific Northwest National Laboratory, Richland, WA, 99352, USA.

*Correspondence to*: Ye Liu (ye.liu@pnnl.gov) and Yun Qian (yun.qian@pnnl.gov)

**Abstract.** We investigate the sensitivity of turbine-height wind speed forecast to initial condition (IC) uncertainties over the Columbia River Gorge (CRG) and Columbia River Basin (CRB) for two typical weather phenomena, i.e., local thermal gradient induced marine air intrusion and a cold frontal passage. Four types of turbine-height wind forecast anomalies and their associated IC uncertainties related to local thermal gradients and large-scale circulations are identified using the self-organizing map (SOM) technique. The four SOM types are categorized into two patterns, each accounting for half of the ensemble members. The first pattern corresponds to IC uncertainties that alter the wind forecast through modulating weather system, which produces the strongest wind anomalies in the CRG and CRB. In the second pattern, the moderate local thermal gradient and large-scale circulation uncertainties jointly contribute to wind forecast anomaly. We analyze the cross-section of wind and temperature anomalies through the gorge to explore the evolution of vertical features of each SOM type. The turbine-height wind anomalies induced by large-scale IC uncertainties are more concentrated near the front. In contrast, turbine-height wind anomalies induced by the local IC thermal uncertainties are found above the surface thermal anomalies. Moreover, the wind forecast accuracy in the CRG and CRB are limited by IC uncertainties in a few specific regions, e.g., the 2-m temperature within the basin and large-scale circulation over the northeast Pacific around 140°W.

## 1 Introduction

Renewable energy have become an alternative to fossil fuels in the last few decades (Al-Dousari et al., 2019) and wind-generated electricity has seen huge growth worldwide (Shaw et al., 2019). It provided more than 8.7% of Unite States electrical power production in 2020 (https://www.eia.gov/electricity/monthly), ranking as the top renewable energy source (National Renewable Energy Laboratory, 2008; Shaw et al., 2009; Oren, 2012; Willis et al., 2018). To successfully manage wind energy, it is of immense importance to accurately forecast the power supplied to the power grid (Siuta et al., 2017a), which relies on the performance of numeral weather prediction (NWP) models in representing the flow features (Siuta et al., 2017b; Willis et al., 2018; Smith and Ancell, 2019)

A major challenge in forecasting wind power generation using NWP models is the significant variability in the planetary boundary layer (PBL) flows (Smith and Ancell, 2017). In the western United States wind farms are normally located in areas with complex terrain, introducing additional complexity to wind forecasting. The performance of NWP models is sensitive to



resolution, model physics, initial/boundary conditions, and parameterizations of the sub-grid processes (e.g., Yang et al.,
      2013, 2017, 2019; Qian et al., 2015; Siuta et al., 2017a; Berg et al., 2019, 2021; Smith and Ancell, 2019), which ultimately
      influences forecasts of wind power (Banta et al., 2013, 2018). The uncertainties associated with the model initialization
      decreases the reliability of the deterministic forecasts and can have serious financial consequences (Marquis et al., 2011). In
      an operational mesoscale model, the initial conditions (ICs) are frequently provided either by global NWP model analysis

and forecasts, or limited area models. Forecast uncertainties may develop from limitations in the ICs, due to their coarse grid
      spacing and/or the lack of sufficiently dense data sources, especially over the ocean. Deppe et al. (2013) performed a
      sensitivity study using ensembles of PBL schemes and IC sources. They found that the perturbation of IC produced a larger
      ensemble spread than application of different PBL schemes, indicating that changes (errors) in IC can lead to large
      uncertainties in wind flow forecasts. Smith and Ancell (2019) further reported considerable differences in the results of

parametric sensitivity among the ensembles started from different ICs. By characterizing NWP model forecasting errors
      using doppler-lidar measurements over the Columbia River Basin, Banta et al. (2020) suggested that the forecasting errors
      could be imported from upstream or generated locally at an earlier time, which emphasized the impacts of both local thermal
      gradient and large-scale circulation initial status on wind forecast. By linking the forecasting errors and IC uncertainties,
      Smith and Ancell (2017) successfully identified the optimal target regions where deployment of additional observations

would improve wind forecasts.
      The ensemble sensitivity analysis (ESA) is an efficient approach to link a forecast scalar, such as the maximum wind speed,
      to atmospheric variables at an earlier time. This approach utilizes a suite of ensemble simulations and has been widely used
      to assess the impact of IC uncertainties on forecasts (Torn et al., 2006, 2017; Ancell and Hakim, 2007; Hill et al., 2016;
      Smith and Ancell, 2017). The forecast scalar is usually calculated based on the temporal mean or maximum wind speed at a

specific location or averaged over an area from each ensemble simulation. Then the linear regression coefficient between the
      forecast scalar and an initial variable is calculated to assess the impact of changes in this initial variable. Zack et al. (2010)
      applied ESA to wind power forecasts in California. They reported that the forecasting errors of wind speeds at 80-m above
      ground are sensitive to the localized atmospheric features at specific locations of a wind farm. However, responses of wind
      speed forecast to IC uncertainties can vary with time and space, which cannot be assessed using a single forecast scalar.

Smith et al. (2017) defined four forecast scalars and found that the forecast uncertainties in terms of the timing and
      horizontal extent of the wind ramp were sensitive to different IC variables. The self-organizing map (SOM), developed by
      Kohonen (2012), is one of the most popular nonlinear pattern recognition techniques. This methodology allows us to identify
      different kinds of wind speed patterns according to the spatial distribution and temporal evolution of wind anomalies induced
      by IC uncertainties.

The Columbia River Gorge (CRG) and Columbia River Basin (CRB) in the Pacific Northwest of the United States are
      characterized by consistent high winds, and have a large number of wind farms and over 6000 megawatts of installed
      capacity (Shaw et al., 2019; Wilczak et al., 2019; Banta et al., 2020). This region includes the Cascade Mountain Range,
      which is cut through by the CRG, linking the Pacific coastal plains with the inland CRB. In a typical warm season, a





subtropical ridge over the eastern Pacific intensifies and moves northward, which results in a higher surface pressure
building up and moving towards the Pacific Northwest coast (Baker et al., 1978), leading to strong westerly winds in the
gorge. In addition, the warm season pressure gradient across the gorge is enhanced by the thermal low that often develops
over the hot interior. The maximum surface temperature over the CRB often exceeds 35 °C in the afternoon (Banta et al.,
2020). Sharp wind ramps are frequently associated with a strong thermally driven gradient, with the total aggregated
normalized power generation fluctuating from near zero to 100% (Wilczak et al., 2019). The local thermal gradient-induced
diurnal wind patterns are often interrupted by synoptic-scale systems, such as strong cold-fronts (Sharp and Mass, 2004),
leading to strong wind ramps (Berg et al., 2021). The combined effects of various weather systems and a general lack of
observations over the Pacific Ocean increase the complexity of wind forecast and make the forecasting errors sensitive to the
atmospheric ICs over the Pacific Northwest.

This study aims to 1) assess the sensitivity of turbine-height wind speed forecasts in the CRG and CRB to IC uncertainties
related to local thermal conditions and large-scale circulations, and 2) identify the regions of IC uncertainties which have the
largest influence on wind forecast. In particular, we focus on two typical summer weather systems associated with a local
thermal gradient and large-scale cold frontal passage. Extensive efforts have been made to develop scale-adaptive physical
parameterizations and parameter optimization during the Second Wind Forecast Improvement Project (WFIP2; Olson et al.,
2019b; Shaw et al., 2019; Wilczak et al., 2019), but there has been less attention to the impact of ICs on the forecast
skill (Smith and Ancell, 2019). The WFIP2-optimized NWP model is used in this study. We use ESA to generate ensemble
wind forecasts from perturbed ICs. We then introduce the SOM to classify forecast anomalies and associated IC
uncertainties. This paper is organized as follows: Section 2 introduces the case selection, experimental setup, and methods.
Section 3 discusses the main results, including the common features of simulated wind speed anomalies induced by IC
perturbation, four types of SOM wind speed patterns and their associated ICs, and the spatial distribution and temporal
evolution of the forecasting uncertainties in different SOM types, including their vertical characteristics. Discussions and
conclusions are given in Section 4.

## 2 Experimental setup and method

### 2.1 Case selection and synoptic condition of case study

We selected two cases with typical weather systems in this area to assess the impacts of IC uncertainties related to local
thermal conditions and large-scale circulations, represented using the 2-m temperature, and 500-hPa geopotential height and
surface pressure, on the wind forecast in the CRG and CRB. Both cases occurred during the WFIP2 field campaign. In the
first case, August 16-17, 2016, the local thermal-driven pressure gradient produced diurnally varying westerly winds through
the gorge in the absence of strong synoptic forcing (Shaw et al., 2019). The flow characteristics were documented by Banta
et al. (2020) using lidar-measured wind speed profiles taken across the CRB. The thermal-driven acceleration of the wind
was observed in the late afternoon, exceeding 12 m s⁻¹ at night over the turbine-height, then gradually slowing. This case is



hereafter referred to as the sea-breeze case. In the second case, August 21-22, 2016, a synoptic-scale cold frontal passage was observed in the CRB, driving cross-barrier flows over the Cascade Mountain Range. This period was marked by a wind ramp event with a maximum speed reaching 14 m s⁻¹ in the CRG (Berg et al., 2021). We denote it hereafter as the cold-front case.

**2.2 Initial condition perturbations**

The Advanced Research version of the Weather Research and Forecasting Model (WRF-ARW) version 4.2 (Skamarock et al., 2019) is used. The model configuration consists of an outer domain D1 with 36-km horizontal grid spacing and an inner domain D2 with 12-km grid spacing (Fig. 1). The model includes 55 vertical layers with 10-27 model layers are found in the planetary boundary layer and approximately 15 m vertical interval within 200 above the surface. The lateral boundary
conditions (LBCs) for D1 are from North American Regional Reanalysis (Mesinger et al., 2006). The LBCs for D2 are provided by D1 using one-way nesting. A standard set of model physics parameterization is used in both domains, including the MYNN-EDMF PBL scheme (Nakanishi and Niino, 2006, 2009; Olson et al., 2019a), the MYNN surface layer scheme (Dyer and Hicks, 1970), Rapid Radiative Transfer Model (RRTMG) long- and short-wave radiation schemes (Iacono et al., 2008), and the aerosol-aware Thompson microphysics scheme (Thompson and Eidhammer, 2014). The Rapid Update Cycle
(RUC) land surface model (Smirnova et al., 2016) is used to represent surface processes.

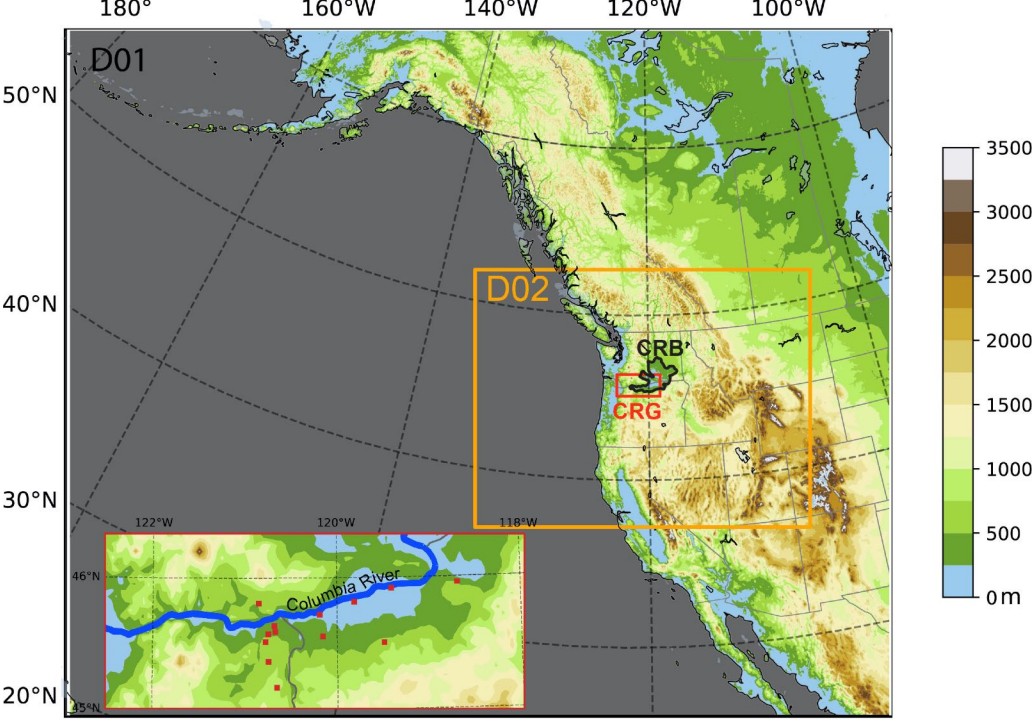

**Figure 1. Model domain with terrain elevation indicated by colors. The Columbia River Gorge (CRG) and Columbia Basin (CRB) are outlined in red and black, respectively. The insert axes amplify the CRG with WFIP2 sodar and lidar locations.**





For each case, a suite of 100 ensemble members is initialized from NARR forecasts perturbed using the climatological covariance in the WRF data assimilation system (WRFDA). Following the routine used in Smith and Ancell (2017), the ensemble members are cycled for a 48-h spin-up period to allow for the development of flow-dependent covariances. Assimilation is performed every 6-h during the spin-up period on the outer domain. After the spin-up, a 48-h extended forecast is conducted with the boundary conditions provided every 3 hours by NARR. The only difference between the members in the extended forecast is their initial conditions. The starting hour of the extended forecast is denoted as 0 h. The first 6-h of the forecast is regarded as a spin-up. Therefore, uncertainties of meteorological conditions at 6 h are referred to as IC uncertainties in the following analyses.

## 2.3 SOM analysis

The SOM is a clustering method that projects high-dimensional data to a visually comprehensible, two-dimensional map. It provides a spatially organized set of patterns of data variability and has been widely used in atmospheric sciences (e.g., Ohba et al., 2018; Song et al., 2019; Spassiani and Mason, 2021). In this study, we apply SOM to recognize patterns of wind speed forecast anomalies and corresponding IC uncertainties. The turbine-height (80-m) wind vectors from D2 over the CRB from each ensemble member are used. The wind vectors are collected every 3 hours during the 24-h time window (from 24 to 48 h) from each grid point and are used as the input vectors for SOM. In this way, the spatial distribution and temporal evolution of the wind vectors are counted during SOM clustering. The input vectors are normalized by removing the regional mean and divided by their standard deviations at each timestep. During the training phase, the initial nodes for SOM clustering are assigned randomly or more efficiently, as used here, selected from the leading empirical orthogonal functions (EOFs). Then the input vectors are mapped to nodes with the closet prototype vector using a Euclidean distance measured between the input vector and the initial nodes. The best-matching node (or the winner node) is the one with the smallest distance. Then the winner nodes are adapted to represent the distribution of the data better, and a neighborhood function is applied to determine the strength of adaptation of the adjacent node. The choice of prescribed SOM node number is a trade-off between distinctiveness and robustness. We test six-nodes SOM clustering for the sea-breeze (Fig. S1 in the supplementary material) and cold-front (Fig S2) cases. The results show that some nodes are redundant. Therefore, we choose four nodes for both cases, which allow us to capture distinct patterns of forecast wind speed while minimizing redundant nodes that are similar. The abovementioned SOM clustering is based on basin-wide turbine-height wind vectors over the CRB. Besides, we also conduct SOM clustering using WRF-simulated wind speed taken at grid points closest to the 13 observation sites located over the CRG. The two SOM clustering results will be compared to discuss the robustness of our findings in Section 4.



## 3 Results

### 3.1 Common features of the simulated winds and associated initial conditions

Our analysis starts with introducing the meteorological background during the two cases, followed by the common features associated with wind forecast anomalies induced by IC uncertainties.

For the sea-breeze case, a weak-large scale forcing is observed over the Pacific Northwest, consisting of a deepened Aleutian Low and weakened North American trough, a slightly southwest-northeast tilted westerly jet, and eastward extension of the subtropical high (Fig S3). The daytime temperature exceeds 35 °C in the CRB and is about 10 °C warmer than the coastal

areas. The thermal-driven flows dominate the CRG and CRB and have distinct diurnal variations. The CRG wind speed indicates that the winds reach peak speed near 2100 hours local time, weakening the following morning (Fig. 2a). There is generally good agreement between the observed and simulated wind speed across the CRG for the first 30 h (of 48 h period), but the simulated wind speed decreases quickly after that. The peak wind speed (observed at forecast hour 29) features strong westerly winds east of the Cascade Mountains and across the gorge (Fig. 2c).

To assess the common features of the wind forecast anomalies induced by IC uncertainties, we divide the ensemble members into two sets based on whether their basin-averaged turbine-height wind speed is greater or less than the ensemble mean averaged over 24-48 h. The wind speed differences between the two sets are mostly found within the basin, with westerly components across the gorge (Fig. 2e). The wind differences are associated with a dipole change in 500-hPa geopotential height (GPH), increasing over the southwest United States and the adjacent Pacific and decreasing over the British Columbia

and coastal areas at 6 h (Fig. 3a). The pattern aloft suggests that the eastward extension of the subtropical high and the southward displacement of the 500-hPa jet at 6 h tend to accelerate wind speed in the CRB during hours 24-48 of the forecast. The wind speed difference is also associated with enhancement of surface low (Fig. 3b) and warming (Fig. 3c) over the northwest United States at 6 h. This anomalous surface pattern has been observed to enhance the sea-breeze forcing and turbine-height wind speed (Wilczak et al., 2019).



**Figure 2. (a)** Time-series of observed and simulated turbine-height wind speed computed across the site locations, as well as time-series of the simulated range between the 2.5 and 97.5 percentiles of ensemble members, **(c)** ensemble mean wind speed at 29 h,





**and (e) composite differences between two sets with wind speed greater or less than the ensemble mean at 29 h for the sea-breeze**
**case. (b, d, f) are the same as (a, c, e), respectively, but for the cold-front case and (d, f) at forecast hour 35. Gray lines in (c-f)**
**indicate terrain elevation of 500 (thin) and 1000 m (thick). Wind vectors are shown when either the zonal or meridional wind**
**anomalies are greater than 5 ms-1 (c-d) and significant at the 0.05 level (e-f), respectively.**

For the cold-front case, the North American trough is deeper than the sea-breeze case, and the subtropical high extends
northward, tilting the westerly jet toward a northwest-southeast orientation (Fig. S4). The surface is characterized by an
offshore ridge and an inland low-pressure center. The CRG turbine-height wind speed ranges from 2 to 15 m s$^{-1}$, peaking at
forecast hour 35 (Fig. 2b). The model simulations capture the wind ramp event, but the simulated increase in wind speed is
smaller and occurs later than is observed. The underestimation also exists in our previous study using a small domain for the
same location and period but is smaller in magnitude than the current analysis (Berg et al., 2021). Therefore, we assume that
forecasting uncertainties from the outer domain may cause underestimation of the wind speed. The common features for the
cold-front case are very different from those in the sea-breeze case. Wind speed differences in the cold-front case broadly
spread over the Pacific Northwest and offshore regions (Fig. 2e). The acceleration of the turbine-height wind is associated
with a positive 500-hPa GPH anomaly over the northeast Pacific Ocean and a negative one to the southwest of it at 6 h (Fig.
3d). A decrease in 500-hPa GPH is also found coastal Canada. The pattern aloft suggests a further northeastward
displacement of subtropical high towards the coast and a clockwise tilting of the westerly jet towards the basin. In response
to IC anomalies aloft, a similar pattern is found in surface pressure (Fig. 3e), where the pressure gradient intensifies between
the northeast Pacific Ocean and Northwest US. The large-scale pattern accelerates wind speed over 24-48 h in the northwest
US. Meanwhile, the moderate anomalous warming over the east Cascade Mountains at 6 h enhances the local thermal
gradient, which also facilitates the subsequent wind speed increase (Fig. 3f).

The common feature analyses suggest that the wind speed anomalies over the CRG and CRB in both sea-breeze and cold-
front cases are linked to IC uncertainties. The sea-breeze case features wind speed anomalies that are mainly within the basin
and a thermally induced surface pressure gradient along the coast at the initial hour. In contrast, anomalous winds are found
broadly over the whole Pacific Northwest in the cold-front case, with large anomalies in the 500-hPa heights and surface
pressure leading to impacting large-scale circulation.



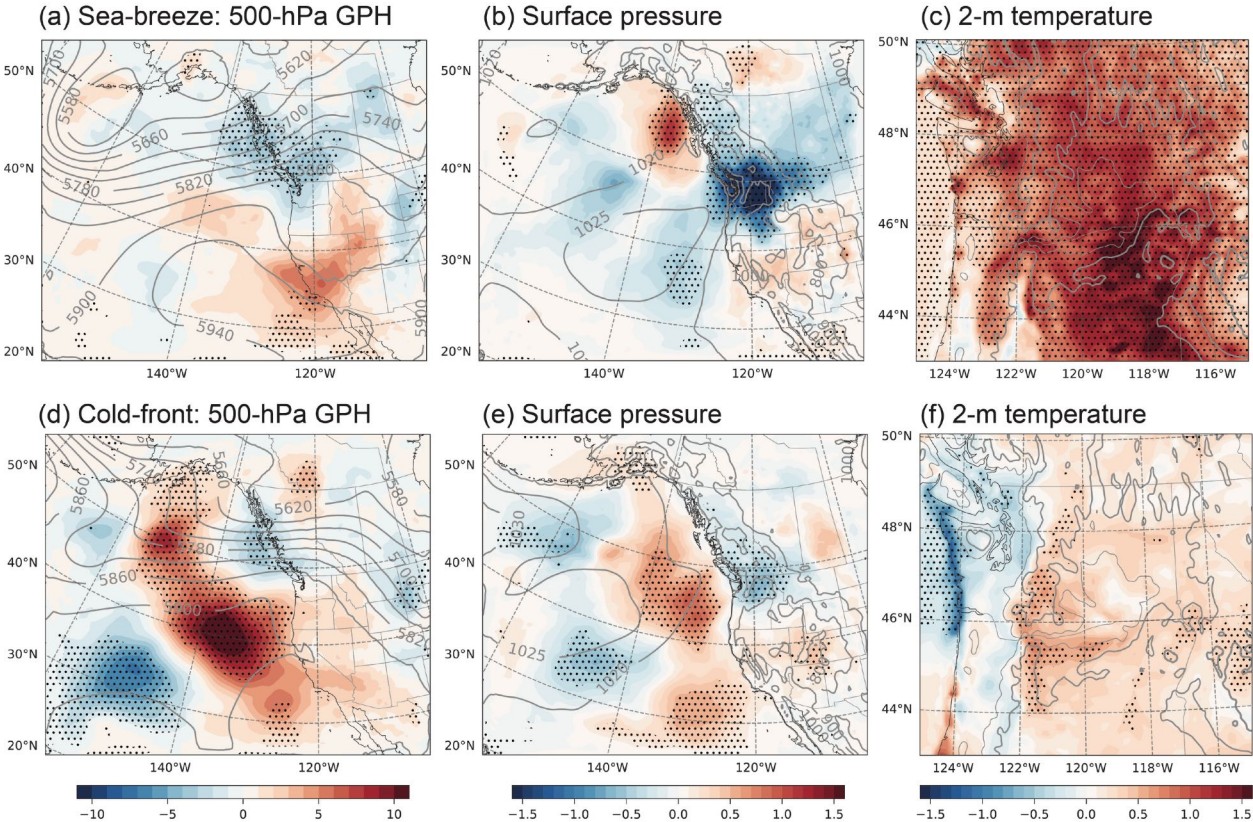

**Figure 3. Composite differences in (a) 500-hPa GPH, (b) surface pressure, and (c) 2-m temperature for the sea-breeze case at forecast hour 29. (d-f) are the same as (a-c) except for the cold-front case and at forecast hour 35. The areas with dots indicate the differences are significant at the level of 0.05. The contours in (a-b and d-e) are ensemble mean at initial hour. Gray lines in (c, f) indicate terrain elevation of 500 (thin) and 1000 m (thick).**

### 3.2 Four types of anomalous turbine-height wind speed and their associated initial conditions

We utilize SOM, which considers the spatial distribution and temporal evolution of wind speed forecasts, to categorize the patterns of wind anomalies into four types using wind vectors over the basin during 24-48 h. The mean of wind speed for each SOM type generally resembles the ensemble means shown in Fig. 2c-d, yet there are differences in the magnitude of wind speed between different types. Types are sorted by wind speed, so that type 1 has the strongest winds across the gorge and basin; Types 1 and 2 generally correspond to the larger wind speeds, while types 3 and 4 correspond to the weaker winds.

During the sea-breeze case, the first SOM type is marked with persistently strong winds, with maximum anomalies occurring at 29 h (about 1.5 m s$^{-1}$, Fig. 4). Strong westerly winds are found in the CRG and CRB (Fig. 5a). Positive and negative 500-hPa GPH anomalies are found at 6 h in the southwest United States and British Columbia, respectively (Fig. 6a). At the same time, a strong 2-m temperature warming east of the Cascade Mountains enhances the local thermal gradient, strengthening the thermal low over the basin. The enhanced inland warming is the primary driver that accelerates westerly winds through





the sea-breeze mechanism (Fig. 6e, i). Type 4 shows opposite signals to type 1 in anomalous wind speed and direction, 500-
        hPa GPH, surface pressure, and temperature (Fig. 5d and Fig. 6d, h, l). Types 1 and 4, accounting for 50% of the ensemble
        members, are consistent with the common features associated with turbine-height wind forecasting anomaly (Fig. 2). These
        two types represent the impact of IC uncertainties that alter the weather system.

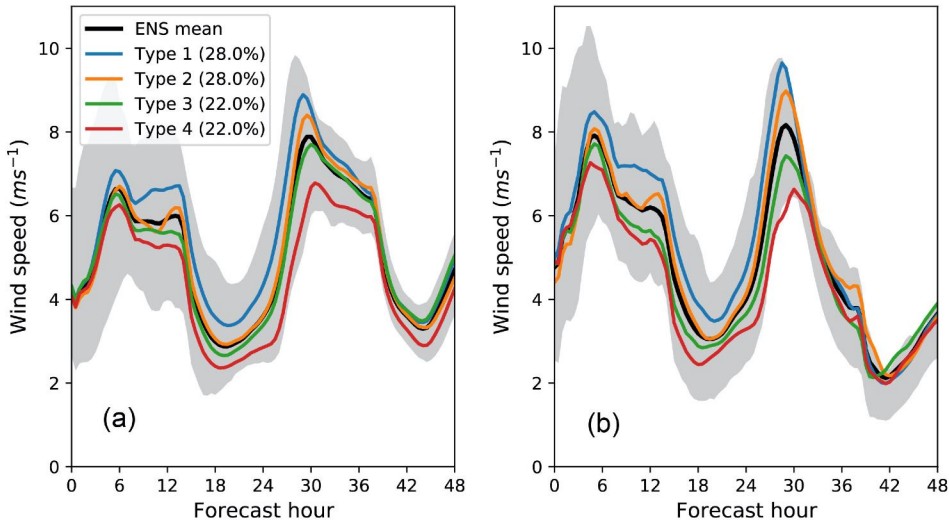

**Figure 4. Time-series of turbine-height wind speed during the sea-breeze case averaged (a) over the basin and (b) across the gorge.**
**The solid black line indicates the ensemble mean, while the colored lines indicate the mean of each SOM type. The envelope shows**
**the 2.5th and 97.5th percentiles of ensemble members. Numbers in parentheses indicate the percentage of each type.**

        For type 2, we also find an enhanced inland heating at 6 h, which is weaker than that in type 1 and causes a non-significant
        surface low over the CRB (Fig. 6f, j). Meanwhile, a significant positive surface pressure anomaly establishes over the

northeast Pacific Ocean around 140°W, although there is no significant change in 500-hPa GPH (Fig. 6f). This large-scale
        anomaly and enhanced inland heating jointly contribute to intensifying surface pressure gradient over the Pacific Northwest,
        highlighting the importance of uncertainties in local thermal gradient and large-scale circulation. As a result, a moderate
        wind speed acceleration is found in type 2 (Fig. 5b), with a maximum of 0.7 m s$^{-1}$ increase at 29 h. Different from type 1,
        wind speed accelerations in type 2 are only found in the CRG and downwind of the Cascade Mountains. Type 3 has general

opposite signals to type 2. An anomalous low-pressure center is found over the northeast Pacific Ocean, accompanied by a
        moderate cooling (non-significant with warming spots) in the basin. Those anomalies are responsible for wind speed
        decrease in type 3 by decreasing pressure gradient over the Pacific Northwest. We find that changes in 500-hPa GPH tend to
        weaken the westerly jet, which may also contribute to negative wind anomaly over the basin.  Types 2 and 3 correspond to
        the combination of moderate uncertainties in initial local thermal gradient and large-scale circulation, which leads to distinct

anomalous flows across the gorge.



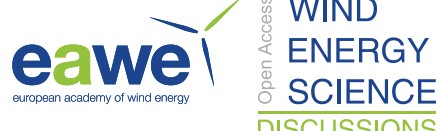

**Figure 5.** Turbine-height wind speed differences from the ensemble mean of each SOM type for the sea-breeze case at 29 h. Wind vectors are shown when either the zonal or meridional wind anomalies are significant at the 0.05 level. Colors indicate the wind speed anomaly. Gray lines indicate terrain elevation of 500 (thin) and 1000 m (thick).



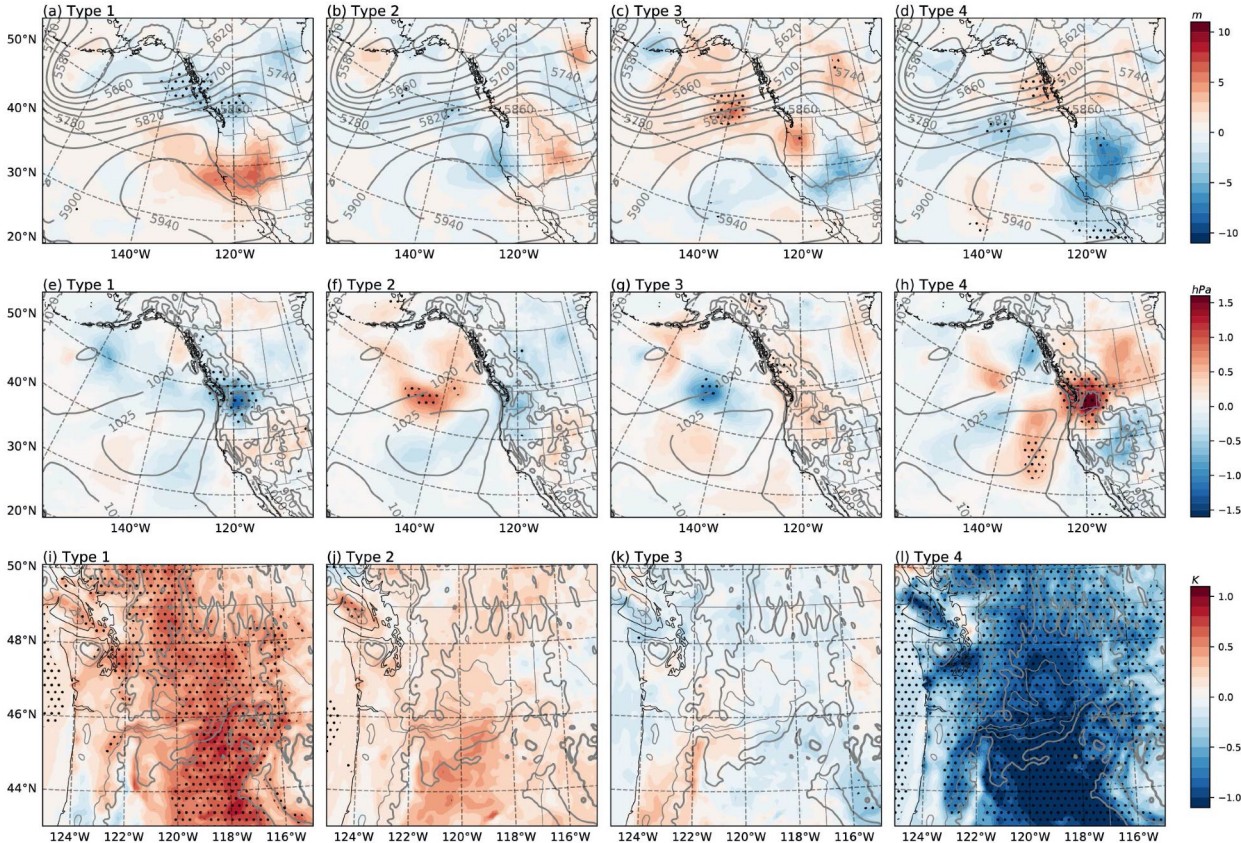

**Figure 6. (a-d) 500-hPa geopotential height, (e-h) surface pressure, and (i-l) 2-m temperature anomaly for each SOM type from ensemble average for the sea-breeze case at 6 h. Contours in (a-h) indicate the ensemble average. Gray lines in (i-l) indicate terrain elevation of 500 (thin) and 1000 m (thick). The dots indicate the differences are significant at the level of 0.05.**

During the cold-front case, type 1 corresponds to persistently strong winds over the simulation period (Fig 7), with enhanced westerly winds found across the Cascade Mountains and east of the CRB, as well as enhanced northerly winds offshore (Fig 8a). Those anomalous winds are associated with dipole anomaly centers at 500-hPa GPH in the northeast Pacific, which suggests a northeast displacement of the subtropical high (Fig. 9a). At the surface, a significant surface pressure increase is found offshore, indicating a strengthening of the surface pressure gradient. The large-scale changes in 500-hPa GPH and surface pressure in type 1 are both conducive to cold-front development. In contrast, type 4 features persistently weak winds (Fig. 7b), anomalous easterly flows east of the Cascade Mountains and southerly flows offshore (Fig. 8d), and large-scale ICs that are unfavorable to cold-front development (Fig. 9d, h). Consistent with our findings in the sea-breeze case, types 1 and 4 here represent the impact of IC uncertainties that alter the weather system. Yet during the cold-front case, strong wind anomalies associated with types 1 and 4 are primarily driven by large-scale IC anomalies, while they are mainly caused by local thermal gradients during the sea-breeze case.





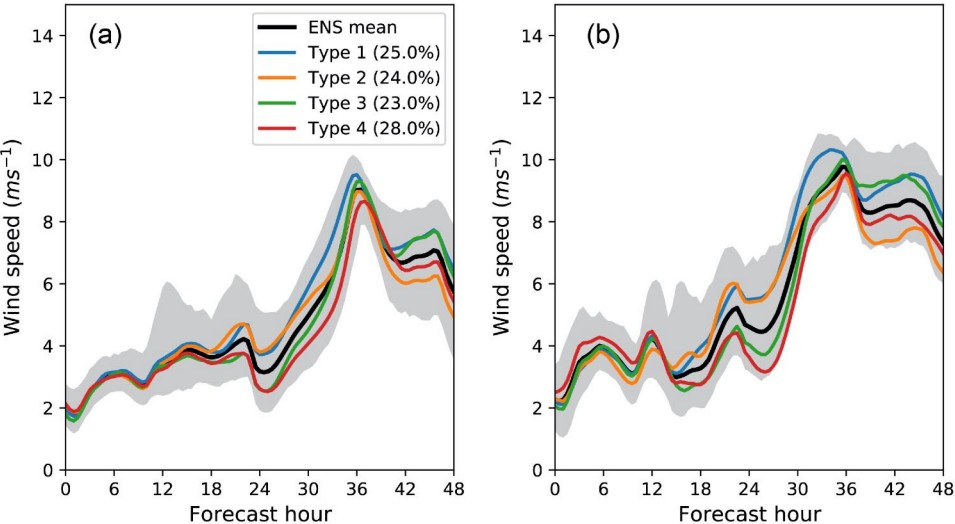


**Figure 7. The same as Figure 4, but for the cold-front case.**

In type 2, the wind acceleration is comparable to type 1 during 18-30 h (Fig. 7b). After that, wind speed quickly decreases and becomes smaller than the ensemble mean. There is a small and non-significant wind speed anomaly over the basin at forecast hour 35 (Fig. 8b). The wind acceleration before 30 h is associated with an eastward extension of subtropical high

and a moderate enhancement in the surface pressure gradient over the Pacific Northwest (Fig. 9b, f), which are similar to, but weaker than, type 1. The large-scale IC anomalies, accompanied by weak inland heating (Fig. 9j), lead to marine intrusion flows into the basin. The cool and moist intrusion flows cause a reversed temperature gradient, which surpasses the large-scale forcing and results in the transition from positive to negative anomalies in wind speed at 30 h (discussed further in section 3.3). In contrast to type 2, type 3 generally corresponds to large-scale anomalies that are unfavorable to the cold-front

development and a colder interior (Fig. 9c, g, k). We find an opposite transition from negative to positive anomalies in wind speed before and after 30 h in type 3 (Fig. 7).





**Figure 8.** The same as Figure 5, but for the cold-front case



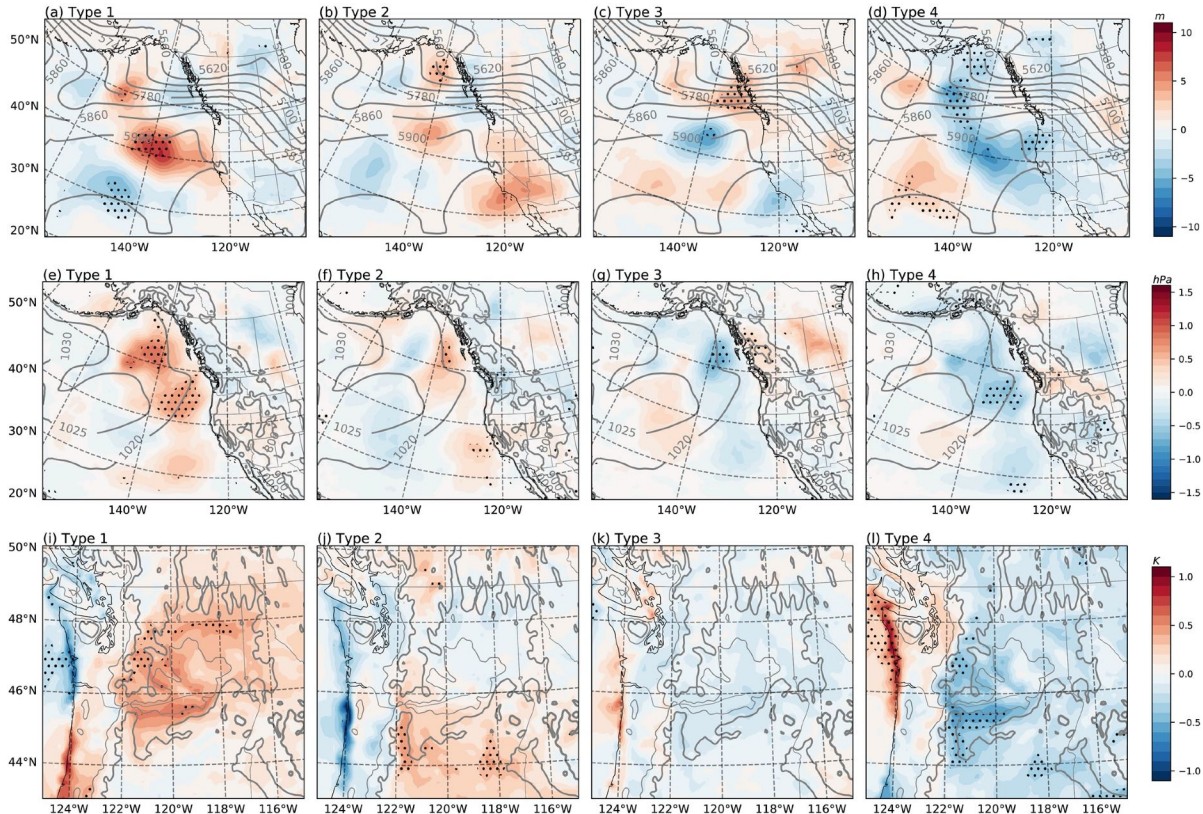

**Figure 9. The same as Figure 6, but for the cold-front case.**

## 3.3 Temporal evolution of wind speed

The IC uncertainties related to both local thermal gradient and large-scale circulation cause different types of responses in forecasted wind speed. Here we explore how the impact of IC uncertainties on wind anomalies evolve with time. We examine the development of 500-hPa GPH and surface pressure to assess the changes of large-scale circulation while we utilize the cross-section along the gorge (from 45°N, 130°W to 45.6°N, 112°W) at different forecast hours to assess the changes of local thermal conditions and wind speed.

During the sea-breeze case, we do not find a major evolution in the large-scale circulation anomalies in types 1 and 4. In type 1, the dipole changes in 500-hPa GPH are found before forecast hour 12, with positive and negative anomaly centers establishing over the California-Nevada and British Columbia, respectively (Fig. 10a). A dipole is also developed in type 4 before 12 h with opposite signs (Fig. 10d). In both types 1 and 4, the magnitudes of 500-hPa GPH anomalies increase with time, while the locations do not change much, retaining a favorable (type 1) and unfavorable (type 4) large-scale environment for sea-breeze intrusion.



As discussed in section 3.2, the anomalous winds in types 1 and 4 are primarily driven by anomalous inland heating. The changes in wind speed are found at 6 h from the surface through the upper levels (Fig. 11a and d). The wind anomalies

mostly locate above the warming (type 1) and cooling (type 4) areas over the mountains as the PBL grows or collapses (Deppe et al., 2013), along with anomalous sea-breeze air mass across the gorge. The sea-breeze air mass propagates inland during the day and encounters surface heating, forming a sea-breeze front at its leading edge (Simpson, 1994; Banta et al., 2020). During 18-30 h, the sea-breeze front in type 1 advances from the west to the east, and strong wind anomalies pass through the gorge (Fig. 11a, e, i). Meanwhile, surface anomalies establishing over coastal Oregon spread across the barrier

(Fig. 10e, h). The advance of the sea-breeze front simulated by our model corroborates analyses from station observations (Brewer and Mass, 2014) and lidar measurements (Banta et al., 2020). A similar pattern is found in type 4, with anomalous winds in the opposite direction (easterly wind anomaly) driven by extensive inland cooling (Fig. 11d, h, l).

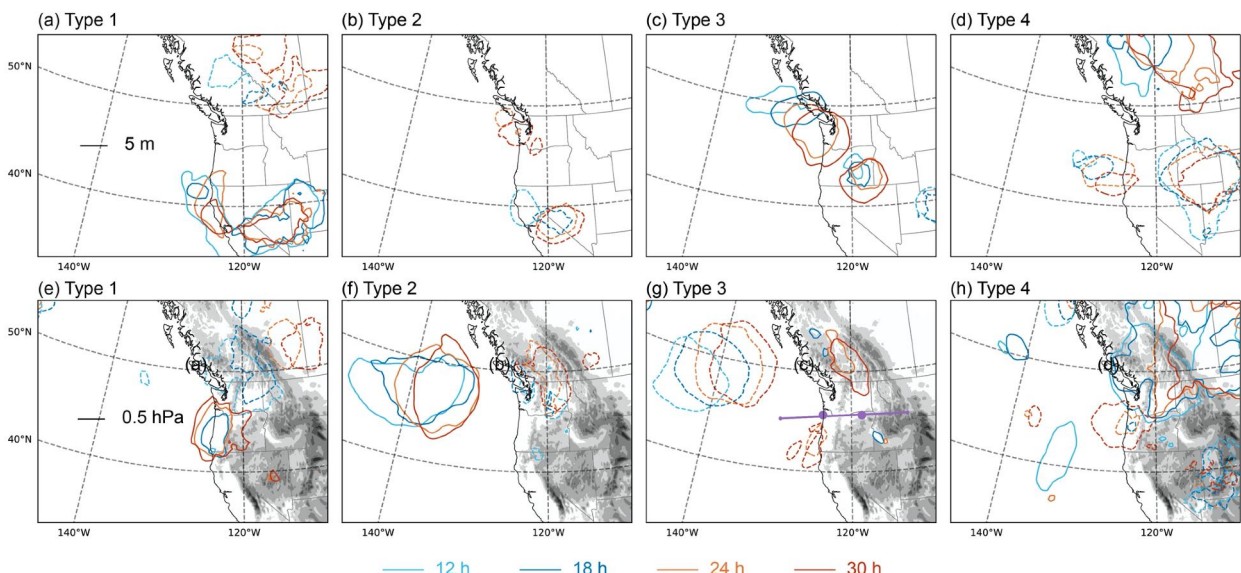

**Figure 10. (a-d) 500-hPa geopotential height anomaly and (e-h) surface pressure anomaly from the ensemble average for each**
**SOM type during the sea-breeze case. The colored lines indicate forecasts at 12, 18, 24 and 30 hours. Dashed contours indicate negative anomaly. The shaded in (e-h) indicates the topography. The location of the cross-section (from 45°N, 130°W to 45.6°N, 112°W) in Figure 11 is marked as the purple line in (g).**

Types 2 and 3 correspond to the joint impact of IC uncertainties associated with both local thermal gradient and large-scale circulations (Fig. 10b, c, and Fig. 11b, c). These types feature weak upper-level anomalies but significant changes in surface

pressure. The surface high (in type 2) and low (in type 3) pressure centers form before 12 h near 140°W and approach the coast during the 18-30 h, accompanied by the thermally driven surface pressure anomalies establishing east of the Cascade Mountains (Fig. 10f, g). From the cross-section view, we find a weak cold front develops offshore at 6 h and advances inland due to both increased surface pressure offshore and inland heating (Fig. 11b, f, j). Type 3 generally has contrasting environmental conditions and opposite wind anomalies to type 2.



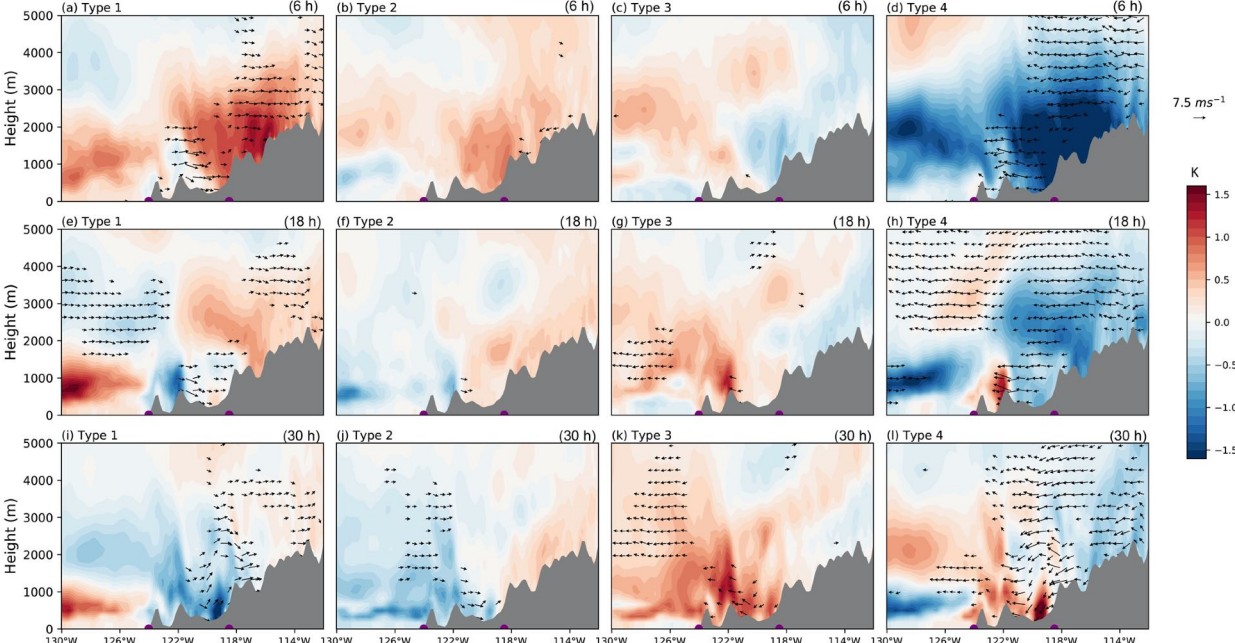

**Figure 11. Cross-section of anomalous wind (vectors) and temperature (colors) at (a-d) 6h, (e-h) 18h, and (i-l) 30h for each SOM type during the sea-breeze case. The gray area indicates the topography. The location of the cross-section is marked in Figure 10.**

During the cold-front case, strong wind speed anomalies in type 1 are primarily driven by large-scale circulation anomalies that are favorable to the cold-front development. The stronger westerly wind anomalies are often observed in the study area and are associated with the intensified offshore ridge at leading hours (Shaw et al., 2019; Banta et al., 2020). In this study, we find strong positive anomalies (10 m) at 500-hPa GPH over the northeast Pacific Ocean strengthen before 18 h and move towards the coast during 18-36 h (Fig. 12a). Meanwhile, a negative anomaly center establishes over the coastal British Columbia before 18 h, which deepens and moves southward to Washington. Similar anomaly patterns are found at the surface. A high-pressure center (1.0 hPa) forms near 140°W and extends toward the coast, intensifying pressure gradient over the Pacific Northwest (Fig. 12e). From the cross-section view, it clearly shows an offshore cold front which coincides with a weak thermally driven front onshore (near 122°W) at 12 h (Fig. 13a). The synoptical-scale front accelerates westerly winds before the front from the surface up to 5 km above ground level (AGL), while the thermally driven front that develops within the basin causes the winds below 1.5 km AGL to accelerate. Along with the inland moving of the large-scale system anomalies, the synoptical-scale front advances eastward and catches up with the thermally driven front at 24 h, which leads to stronger westerly winds in the basin (Fig. 13e). The turbine-height winds peak at around 36 h (Fig. 7a) when the anomalous front passes and leaves behind cold and moist air in the basin (Fig. 13i). At 36 h, anomalous winds develop well from the surface to 500 hPa. By contrast, type 4 corresponds to an unfavorable large-scale initial condition that causes turbine-height wind suppression.

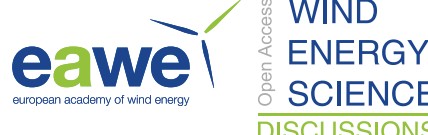

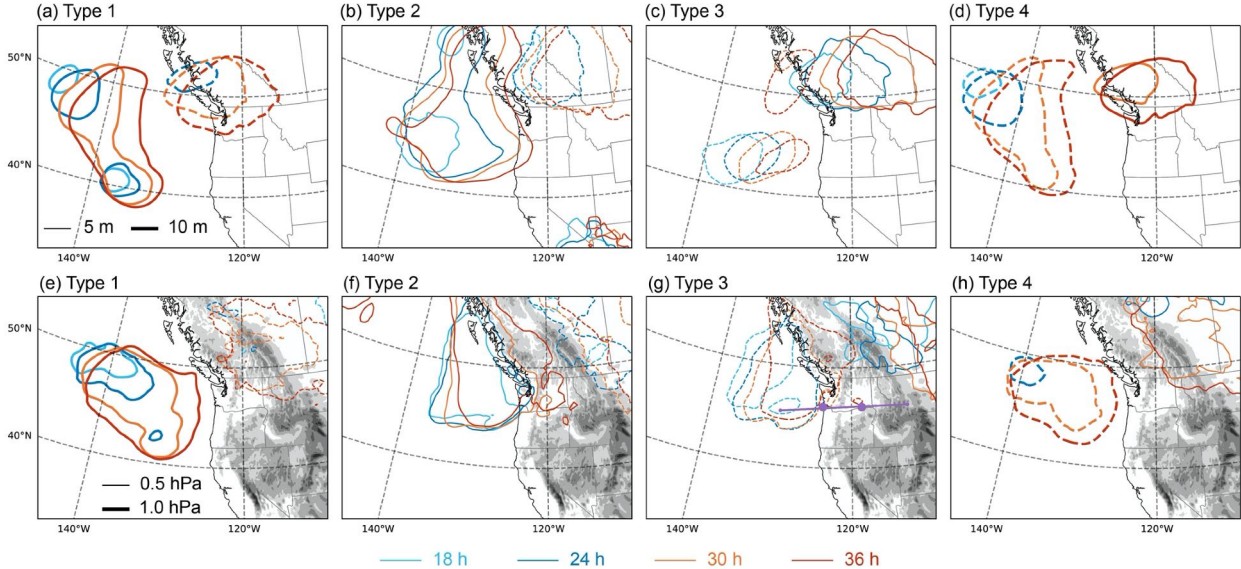

**Figure 12. The same as Figure 10, but for the cold-front case.**

Types 2 and 3 correspond to the eastward displacement of the large-scale system, with smaller magnitudes compared to types 1 and 4 (Fig. 12b, c, f, g). The wind anomalies across the gorge are driven by the joint impact of the anomalous local thermal gradient and large-scale circulation (Fig. 13). In type 2, the enhanced westerly winds bring cool-moist air into the basin before 12 h. The corresponding wind anomaly profiles exhibit features of cold-front acceleration, i.e., strong westerly just ahead of the front. When the cool-moist air cools the basin after 24 h, the reversed thermal gradient surpasses the cold-front forcing, leading to anomalous easterly flows.



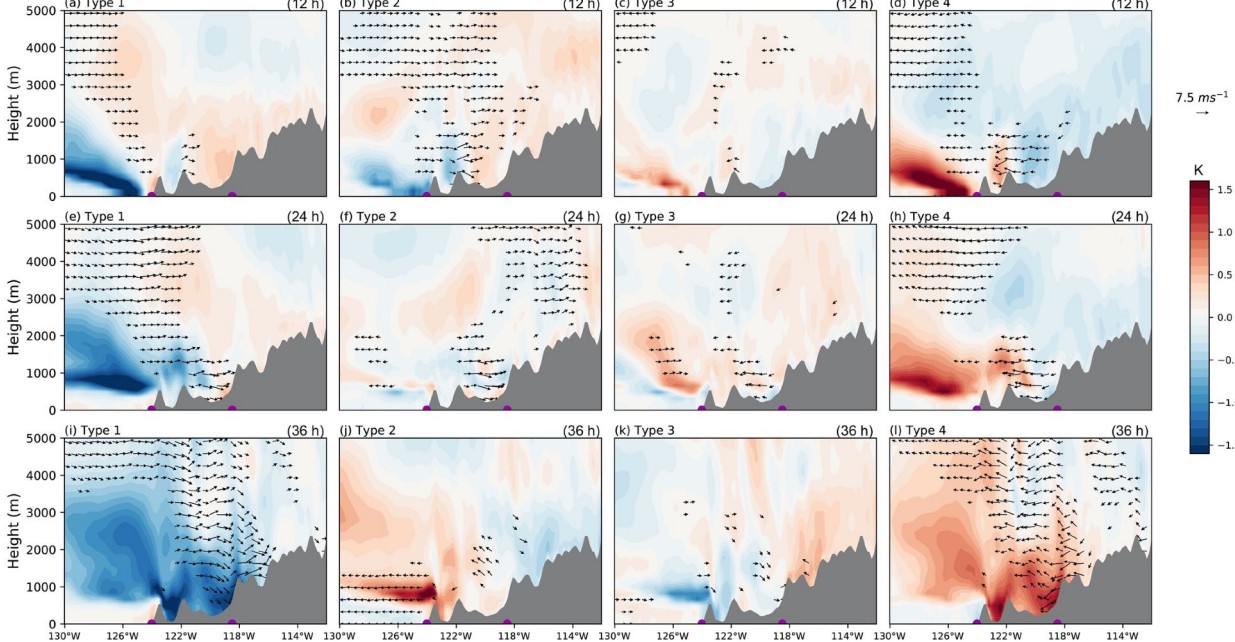

**Figure 13. The same as Figure 11, except for (a-d) 12h, (e-h) 24h, and (i-l) 36h during the cold-front case.**

## 4 Discussion and Summary

During the summer over the Pacific Northwest, the offshore ridge and strong thermal gradients can produce diurnal circulations that develop and interact on various spatial and temporal scales. The diurnal cycles occur concurrently with weak large-scale fronts, troughs, and lines of moist convection and can be interrupted by a strong cold frontal passage. The uncertainties in any of those local and large-scale conditions at an earlier hour can comprise the accuracy of subsequent wind forecasts over the CRG and CRB. This paper assesses the impact of the initial condition uncertainties during two periods

influenced by typical weather systems. We utilize the WRFDA technique to generate initial perturbations based on NARR reanalysis, which is then used to initialize 48-h ensemble forecasts. The SOM technique is applied to characterize the major patterns in wind forecasting anomalies and associated IC uncertainties.

We apply the SOM technique and obtain four types of anomalies based on the basin-wide wind vectors. In both meteorological cases, members with strong wind anomalies over the CRB also have the strongest anomalous flows in the

CRG (Fig. 4 and Fig. 7). We further conduct SOM clustering using wind vectors taken from the WRF grid points closest to the site locations in the gorge (Fig. 1). The CRG-based categorization results are generally consistent with those obtained from the basin-averaged analysis. For the sea-breeze case, 81% of members are categorized into the same types with the SOM clustering based on basin-wide wind vectors. For the cold front case, 75% of members have the same categories. The coinciding members are evenly distributed in each SOM type. Furthermore, the composite analyses and cross-section



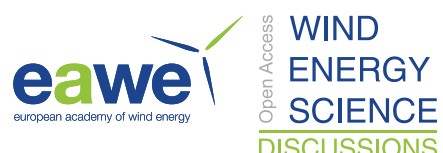

analyses based on the gorge SOM clustering are consistent with those based on the basin-wide clustering. The close connection of wind anomalies in the CRG and CRB indicates that their local and large-scale driving factors are generally consistent, which implies the forecasting anomalies in the CRG and CRB may be caused by IC uncertainties in the same weather systems. The elevation of the CRG is very near sea level, compared to the Cascade passes, whose lowest elevation is 900 m (Sharp and Mass, 2004). Thus, the CRG serves as a primary pathway for the intrusion flows into the basin (Smith and

Ancell, 2019). Observational studies show that flows frequently originate west of the Cascade ranges, propagate through the CRG (Banta et al., 2020), and move eastward through the basin (Brewer and Mass, 2014). Therefore, efforts targeting the improvement of wind forecast in the CRG can also benefit the forecast in the CRB and vice-versa.

The SOM technique efficiently recognizes the common features of anomalous winds and their associated IC uncertainty. In both cases, SOM isolates two types (1 and 4) of wind anomalies corresponding to the strengthening or weakening of the

synoptic weather system. Specifically, during the sea-breeze case, these SOM types feature the condition of the intensified initial thermal gradient across the Cascades, enhanced pressure gradient, and faster winds in the CRG and CRB. In the cold-front case, the anomalous offshore ridge amplifies and approaches the coast, which can be explained by the inland moving of an enhanced cold front that forms offshore.

Besides the IC uncertainties that alter the influencing weather system, SOM also characterizes two types (2 and 3) of wind

anomalies that correspond to the combination of moderate local thermal and large-scale initial uncertainties. These two types of events account for about 50% of the members in both sea-breeze and cold-front cases. Although the anomalous ICs are much weaker than the abovementioned types 1 and 4, they produce comparable wind anomalies, especially over the CRG. This suggests that the local thermal and large-scale initial uncertainties jointly play essential roles in the subsequent wind forecast.

We note that in all types, initial uncertainties in some key regions, such as surface pressure over the northeast Pacific (around 140°W) and surface temperature over the east Cascade, exert significant impacts to the wind forecast in the CRG and CRB in the cases we examine. The knowledge of key regions can be used as a reference for deployment of field campaign, such as in the next stage of the WFIP. The high-quality temperature and pressure observations can be utilized to reduce the IC uncertainties in the wind forecast through the data assimilation system. Note that we mainly focus on IC uncertainties in two

typical weather phenomena in this study. Additional sensitive regions may be needed if we consider other transient synoptic regimes that occur during the summer season (Brewer and Mass, 2014).

**Data Availability**

Data and WRF model configuration used in this manuscript are available from the Atmosphere to Electron Data Archive and Portal at https://a2e.energy.gov/data/wfip2.model/casestudy.wrf.01.seabreeze.01,

https://a2e.energy.gov/data/wfip2.model/casestudy.wrf.02.seabreeze.02,

https://a2e.energy.gov/data/wfip2.model/casestudy.wrf.01.coldfront.01, and



https://a2e.energy.gov/data/wfip2.model/casestudy.wrf.02.coldfront.02. The WRF-ARW v4.2 code is available at
https://github.com/wrf-model/WRF/releases/tag/v4.2.

**Author contribution**

Ye Liu: Methodology, Software, Validation, Formal analysis, Investigation, Data Curation, Writing (original draft, review
and editing).

Yun Qian: Conceptualization, Methodology, Supervision, Formal analysis, Writing (review and editing).

Larry K. Berg: Conceptualization, Methodology, Supervision, Formal analysis, Writing (review and editing), Resources,
Funding acquisition.

The authors declare that they have no conflict of interest.

**Acknowledgments**

This work was supported by the Department of Energy's Wind Energy Technology Office. Computational resources were
provided by Wind Energy Technology Office and the National Renewable Energy Laboratory. The authors thank Dr.
Fengfei Song for providing the SOM scripts, and Dr. Raghavendra Krishnamurthy for reviewing earlier versions of the
manuscript. PNNL is operated by DOE by the Battelle Memorial Institute under contract DE-A06-76RLO 1830.

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
