# Peer review of "Local thermal gradient and large-scale circulation impacts on turbine-height wind speed forecasting over the Columbia Basin"

_Wind Energy Science, 2021_

## Author Response (AR1)

**Response to Reviewer's Comments**

We thank the editors and two anonymous reviewers for their thoughtful and constructive comments and suggestions, which significantly improve the quality of our manuscript. In this revised manuscript, we have tried our best to address all comments and have revised the manuscript accordingly. The reviewer's comments are in black, and our point-by-point responses to the reviewer's comments are in blue.

**Reviewer #1 Evaluations:**

General comment:

Liu et al. have addressed the challenge of weather forecast uncertainties in wind power. The paper with the title "Local thermal gradient and large-scale circulation impacts on turbine-height wind speed forecasting over the Colombian Basin" look into the sensitivity of wind speed forecast to changes in initial conditions related to two weather events: local thermal gradient and the passage of a large-scale cold front. Their work is well written, their work brings attention to the role of initial conditions in the overall wind forecast uncertainty. Although their study area is limited and the result are site-specific they show the importance of considering initial condition (IC) uncertainties in the wind speed forecast for wind power. Their usage of the method self organizing map (SOM) technique clearly illustrates the advantage of this method in terms of clustering and projecting the result onto dominant nodes influencing the forecast uncertainty. I would suggest publication in Wind Energy Science after minor revisions.

R: We greatly appreciate your positive comments. The concerns have been addressed to the extent possible in the revised manuscript.

Specific comments:

RC1) Have the results from the WRF-ARW model run been validated prior to this analysis?

R: Yes, this work is part of WFIP2 project, in which the WRF-ARW model has been validated over many sites within the WFIP2 study area (e.g., Banta et al., 2013, 2017; Q. Yang et al., 2013; B. Yang et al., 2019; Olson et al., 2019). Our previous study (Berg et al., 2021) has evaluated the simulated wind speed using the same model configuration and simulation periods but with an older WRF-ARW version (3.9) and a smaller domain (slightly larger than Washington and Oregon state). The results showed good agreement between the simulated and observed wind speed during both sea-breeze and cold-front influenced periods. We also provided a comparison against observations in this work (Fig. 2a and 2b).

RC2) In the introduction, line 80, the authors write "The WFIP2-optimized NWP model is used in this study.". The authors do not explain this model in any way. Instead, in section 2.2 the authors state that they use the WRF-ARW for the initial condition perturbation. I found this a bit confusing. Please clarify.

R: Sorry for the confusion. We use the WRF-ARW v4.2 which incorporates model development made as part of the WFIP2 project. During WFIP2, a set of model physics schemes were improved, including improvements of planetary boundary layer local and nonlocal mixing, subgrid-scale cloud, wind farm parameterization, and many other aspects (Olson et al., 2019). These updated model components have been shown to improve forecast performance. Some of the updates were implemented in the public WRF-ARW version 4.2 used in this study. We have deleted this sentence (line 80) to avoid confusion.

RC3) Line 46-54: The authors write about ensemble sensitivity analysis (ESA) and self-organizing map (SOM). It would be clarifying if the authors add a line of why they write about those methods. In the end of the paragraph the authors can add something like this: "We use the ESA and SOM to generate..."

R: Point taken. We added the following sentence to the end of this paragraph.

*In this study, we follow the ESA approach to generate a suite of ensemble simulations by perturbing IC and apply SOM analysis to link simulated wind speed features to IC uncertainties.*

RC4) For many of the figures the axis unit or colorbar unit are missing (Figure 1, 2, 3, 5).

R: Sorry about the confusion. The axis and colorbar units are added as suggested.

RC5) In figure 2 the abbreviation "LST" is not explained. Please spell out the abbreviation.

R: "LST" stands for local solar time. We made explanations in the revised manuscript.

RC6) It would also be nice if you set a title in figure 4 and 7 indicating that panel a) corresponds to the Colombia River basin and panel b) is the Colombia River Gorge, and also something in the figure indicating that the figure corresponds to either local gradient or large-scale features. This applies for many of the other figures as well. It would be easier to follow if the figures had a title indicating of the figure is related to the sea breeze or the cold-front case.

R: Per your suggestion, we have added titles to each panel of Figures 4, 5, 7, and 8. We also added case information to all other figures.

RC7) Line 105: Add the abbreviation NARR behind the ``the North American Regional Reanalysis''.

R: Point taken. Thanks.

RC8) Please add the resolution of the reanalysis (NARR).

R: The grid spacing of the NARR analysis is 32 km, which has been added to Lines 106-107.

I found this at NCEP's homepage describing the NARR reanalysis: ``The grid resolution is 349x277 which is approximately 0.3 degrees (32km) resolution at the lowest latitude''. Why have

the authors chosen to run the outer domain (D1) of WRF at an horizontal resolution of 36 km? If the resolution from NARR and from WRF D2 is almost the same, why do the authors use two domains in their model run?

R: In this work, we investigate the impact of large-scale initial conditions (IC) uncertainties over the Pacific Northwest (covered by D1) on the turbine-height wind speed in the CRB. Since NARR reanalysis only provides a single status of IC, we need to generate ensemble simulations by perturbing IC using the WRFDA for D1 region. Starting from NARR reanalysis, we perform assimilation every 6-h on D1 to generate perturbated IC during the 48-h spin-up period.

Besides, our simulation includes large areas of Northwest Pacific to examine the large-scale IC uncertainties. Considering both computational resources and model accuracy, we chose 36 km as the grid spacing of domain 1 which covers large areas in Northwest Pacific, and we chose 12 km grid spacing for inner domain.

RC9) Line 135: The authors write "a neighborhood function is applied". Describe what this is.

R: The neighborhood function is used to adjust the dominant nodes during the training process, which can be demonstrated as follows.

At the $t$th iteration of SOM training process, for a given input vector, the nearest (smallest Euclidian distance) node to the input vector is the best matching node (winner node, $N_c$). Then the winner node and adjacent notes are adjusted toward the input vector. The rate of change $(\Delta d_{ij}(t + 1))$ is given by:

$$\Delta d_{ij}(t + 1) = \alpha(t)h_{ij}^c d_{ij}(t)$$

where $\alpha(t) = 0.85/t$ is a learning rate, $h_{ij}^c$ is the neighborhood function, and $d_{ij}(t)$ is the distance between the input vector and note $N_{ij}$ ($i$ in row and $j$ in column) at the $t$th iteration. The neighborhood function is defined as:

$$h_{ij}^c = \alpha(t) \cdot e^{\frac{-\|R_c - R_{ij}\|^2}{2\left(\eta_{ij}^c(t)\right)^2}}$$

The parameter $\eta_{ij}^c(t)$ represents the radius of the neighborhood to be adjusted, which decreases with $t$. $R_c$ and $R_{ij}$ represent the vector of column and row numbers of $N_c$ and $N_{ij}$, respectively.

In summary, the neighborhood function is a function that determines how many neighborhood nodes around the winner node to be adjusted and the how much to be changed. The adjusted number of neighborhood node decreases with the order number of the current iteration, and the rate of change for a neighborhood node decrease with its distance to the winner node.

We have incorporated the above summary to the revised manuscript Lines 141-144.

RC10) In Fig 2a and 2b, how do the authors explain the large differences between the model and the observed wind speed? (please see RC1).

R: Compared to simulations in the abovementioned previous study (Berg et al., 2021), we used a large outer domain covering the entire northwest Pacific. Therefore, we assume the simulated biases of the outer domain could be the reason for the biases in hub-height wind speed found in the inner domain. Since this work is a sensitive study that assesses the response of hub-heigh wind speed in the CRB to large-scale perturbation, we assume the biases associated with the outer domain simulation should appreciably impact our results. It can be seen from the results that the simulated wind speed spread does not change with the magnitude of bias (Fig. 2a and 2b).

RC11) In caption of Fig 2 the authors write ".. across the site locations". Does the "site locations" refer to the 13 observation sites located over the CRG. Please clarify.

R: Yes, it is clarified in the revised manuscript.

RC12) Line 180: "Wind speed differences in the cold-front case broadly spread over the Pacific Northwest and offshore regions (Fig. 2e)." You refer to Fig 2e) for the cold front case. I believe you mean Fig. 2f)?

R: Yes, it should be Fig. 2f. It has been corrected in the revised manuscript. Thanks for pointing it out.

RC13) Figure 5: label the panels according to the four SOM types: a) Type 1; b) Type 2 etc.

R: Panel title has been added.

RC14) Line 280: The wind anomalies \textit{are} mostly located...

R: Corrected.

RC15) Line 281: "During 18-30 h, the sea-breeze front in type 1 advances from the west to the east, and strong wind anomalies pass through the gorge (Fig. 11a, e, i)." From the figure, I do see the sea-breeze front. However, I would expect the anomalously cold air to extent offshore. The water cools the overlying air, and this air is driven over land by the local thermally driven pressure gradient. Why is the anomalously cold air only located over land? Please explain.

R: Fig. 11 shows the anomaly wind and temperature profile in each type (Type 1 minus ensemble mean). Below we show Figure R1, which is the same as Fig. 11 except that the contours represent mean temperature profile. Type 1 corresponds to sea-breeze front onshore driven by the local thermally driven pressure gradient, and there is cold air extending offshore. In Type 1, we find the temperature increase over land is larger than that over the ocean (Fig. 3c), thus intensifying temperature gradient, accelerating wind speed near the front, and driving additional cold air from the ocean which results in an anomalous cooling after the sea-breeze front (Fig. 11).

[Figure]

**Figure R1. Cross-section of anomalous wind (vectors) and mean temperature (colors) at (a-d) 6h, (e-h) 18h, and (i-l) 30h for each SOM type during the sea-breeze case. The gray area indicates the topography. The location of the cross-section is marked in Figure 10.**

RC16) Line 341-342: The authors write "The CRG-based categorization results are generally consistent with those obtained from the basin-averaged analysis. For the sea-breeze case, 81% of members are categorized into the same types with the SOM clustering based on basin-wide wind vectors. For the cold front case, 75% of members have the same categories."
Where does these numbers come from?

R: The results presented in section 3.2 and 3.3 (Figures 4-12), are associated with the SOM clustering based on the basin-wide vectors (wind speed at each grid point in the basin). We also conducted SOM clustering based on wind vectors taken from the WRF grid points closest to the 13 observation sites in the gorge. The purpose of this analysis is to evaluate if the wind speeds in the basin and gorge are influenced by the same ICs. We calculated the percentage of members that are categorized into the same type using the two SOM clustering approaches. In the sea-breeze case, we found 81 out of 100 members were categorized into the same types using either basin-wide or grid points wind vectors while 19 members were different. The close connection of wind anomalies in the CRG and CRB indicates that their local and large-scale driving factors are generally consistent, which implies that the forecasting anomalies in the CRG and CRB may be caused by IC uncertainties in the same weather systems.

**Reviewer #2 Evaluations:**

This work investigates the sensitivity of turbine-height wind speed forecast to initial condition uncertainties over the Columbia River Gorge and Columbia River Basin using the ensemble sensitivity analysis and self-organizing map (SOM) technique. The paper is well-written, and the analysis is well-presented. It shows the usefulness of SOM in understanding and quantifying the wind forecast uncertainties. However, I do admit that I am not too familiar with this technique and have trouble fully interpreting the results. Nevertheless, I think this work is inspiring to the Wind Energy Science community and is worthy of publication after addressing the following minor issues.

R: We greatly appreciate your positive comments. The concerns have been addressed as possible as we can in this revised manuscript.

Specific comments:

Lines 10-11: why only half of the ensemble members, not the entire ensemble, are used to categorize the SOM type? If the entire ensemble is used, does the pattern of the SOM type change significantly?

R: We used all the 100 members during the SOM clustering and obtained four types of turbine-height wind forecast anomalies. In both sea-breeze and cold-front cases, we find types 1 and 4, and types 2 and 3 showed similar patterns but with opposite signal. For instance, type 1 in the sea-breeze case corresponds to enhanced wind speed associated with strong inland heating at the initial hour, while type 4 corresponds to weakened wind speed associated with weak inland heating. Therefore, the four types can be further categorized into two patterns, each of them accounting for about 50% of the ensemble members.

Line 29-31: A recent NWP study by Xia et al. (2021) might be useful here.

Xia, G., Draxl, C., Berg, L. K., & Cook, D. (2021). Quantifying the Impacts of Land Surface Modeling on Hub-Height Wind Speed under Different Soil Conditions, Monthly Weather Review, 149(9), 3101-3118. https://doi.org/10.1175/MWR-D-20-0363.1

R: The reference has been added to the revised manuscript. Thanks.

Lines 116-117: I believe only a certain number of variables is perturbed. Can you specify that explicitly, rather than just referencing to the reference?

R: Yes. We adopted the WRF data assimilation (WRFDA) technique to generate the ensemble members by perturbing IC during the 48-h spin-up period. The WRFDA is designed to combine observations with an NWP product to provide an improved estimate of the atmospheric state. WRFDA has a built-in method for generating ensemble initial conditions to add random noise to the analysis in control variable space. The control variables used in this study are stream function, unbalanced velocity, unbalanced temperature, unbalanced surface pressure, and

specific humidity. As a result, initial variables including wind speed, temperature, pressure, and specific humidity are perturbed.

Above content has been added to the revised manuscript Lines 118-122.

Line 135: what is the "neighborhood function"?

R: The neighborhood function is used to adjust the dominant nodes during the training processes, which can be demonstrated as follows.

At the $t$th iteration of SOM training process, for a given input vector, the nearest (smallest Euclidian distance) node to the input vector is the best matching note (winner node, $N_c$). Then the winner node and adjacent notes are adjusted toward the input vector. The rate of change ($\Delta d_{ij}(t + 1)$) is given by:

$$\Delta d_{ij}(t + 1) = \alpha(t)h_{ij}^c d_{ij}(t)$$

where $\alpha(t) = 0.85/t$ is a learning rate, $h_{ij}^c$ is the neighborhood function, and $d_{ij}(t)$ is the distance between the input vector and note $N_{ij}$ ($i$ in row and $j$ in column) at the $t$th iteration. The neighborhood function is defined as:

$$h_{ij}^c = \alpha(t) \cdot e^{\frac{-\|R_c - R_{ij}\|^2}{2\left(\eta_{ij}^c(t)\right)^2}}$$

The parameter $\eta_{ij}^c(t)$ represents the radius of the neighborhood to be adjusted, which decreases with $t$. $R_c$ and $R_{ij}$ represent the vector of column and row numbers of $N_c$ and $N_{ij}$, respectively.

In summary, the neighborhood function is a function that determines how many neighborhood nodes around the winner node to be adjusted and how much to be changed. The adjusted number of neighborhood node decrease with the order number of the current iteration, and the rate of change for a neighborhood node decreases with its distance to the winner node.

We have incorporated the above summary to the revised manuscript Lines 141-144.

Please provide figure caption for a), b), c) and d) in Figure 5.

R: Revise as suggested.

Figures 5-6 and Figures 8-9: Does SOM provide variance statistics for each identified pattern (like EOF)? If it does, it is important to provide such information on the figures as well as in the text to facilitate understanding of the relative importance of the identified patterns.

R: The reviewer brings up a good point. To our knowledge, unfortunately, there are no variance statistics that can be applied to rank the relative importance of the identified patterns. The training process of SOM is designed to group similar input vectors together into the same node.

After successful training, the variance of each identified pattern is minimized. Comparison of variance of each pattern does not provide ranking information. The SOM clustering process forms an approximation of the distribution of input data. Therefore, an identified pattern that corresponds to the group with more ensemble members is more likely to happen.

Banta, R. M., Pichugina, Y. L., Brewer, W. A., James, E. P., Olson, J. B., Benjamin, S. G., Carley, J. R., Bianco, L., Djalalova, I. V., Wilczak, J. M., Hardesty, R. M., Cline, J., & Marquis, M. C. (2017). Evaluating and Improving NWP Forecast Models for the Future: How the Needs of Offshore Wind Energy Can Point the Way. *Bulletin of the American Meteorological Society*, *99*(6), 1155–1176. https://doi.org/10.1175/bams-d-16-0310.1

Banta, R. M., Pichugina, Y. L., Kelley, N. D., Hardesty, R. M., & Brewer, W. A. (2013). Wind Energy Meteorology: Insight into Wind Properties in the Turbine-Rotor Layer of the Atmosphere from High-Resolution Doppler Lidar. *Bulletin of the American Meteorological Society*, *94*(6), 883–902. https://doi.org/10.1175/bams-d-11-00057.1

Berg, L. K., Liu, Y., Yang, B., Qian, Y., Krishnamurthy, R., Sheridan, L., & Olson, J. (2021). Time Evolution and Diurnal Variability of the Parametric Sensitivity of Turbine-Height Winds in the MYNN-EDMF Parameterization. *Journal of Geophysical Research: Atmospheres*, *126*(11). https://doi.org/10.1029/2020jd034000

Olson, J. B., Kenyon, J. S., Angevine, W., Brown, J. M., Pagowski, M., & Sušelj, K. (2019). A description of the MYNN-EDMF scheme and the coupling to other components in WRF–ARW. *NOAA Technical Memorandum OAR GSD-61, NOAA*.

Olson, J. B., Kenyon, J. S., Djalalova, I., Bianco, L., Turner, D. D., Pichugina, Y., Choukulkar, A., Toy, M. D., Brown, J. M., Angevine, W. M., Akish, E., Bao, J.-W., Jimenez, P., Kosovic, B., Lundquist, K. A., Draxl, C., Lundquist, J. K., McCaa, J., McCaffrey, K., … Cline, J. (2019). Improving Wind Energy Forecasting through Numerical Weather Prediction Model Development Improving Wind Energy Forecasting through Numerical Weather Prediction Model Development. *Bulletin of the American Meteorological Society*, *100*(11), 2201–2220. https://doi.org/10.1175/bams-d-18-0040.1

Yang, B., Berg, L. K., Qian, Y., Wang, C., Hou, Z., Liu, Y., Shin, H. H., Hong, S., & Pekour, M. (2019). Parametric and Structural Sensitivities of Turbine-Height Wind Speeds in the Boundary Layer Parameterizations in the Weather Research and Forecasting Model. *Journal of Geophysical Research: Atmospheres*, *124*(12), 5951–5969. https://doi.org/10.1029/2018jd029691

Yang, Q., Berg, L. K., Pekour, M., Fast, J. D., Newsom, R. K., Stoelinga, M., & Finley, C. (2013). Evaluation of WRF-Predicted Near-Hub-Height Winds and Ramp Events over a Pacific Northwest Site with Complex Terrain. *Journal of Applied Meteorology and Climatology*, *52*(8), 130503112022000. https://doi.org/10.1175/jamc-d-12-0267.1

---

## Author Response (AR2)

**Comments to the author**:
Dear Dr Liu and co-authors,

I have read your revised manuscript and your response to the reviewers' comments. I think you have addressed all the comments nicely and produced a nearly ready manuscript for publication. I have made a few comments in the document itself (attached). I also noticed that you did not format the references correctly. Would you please read the guidelines in https://www.wind-energy-science.net/submission.html#references and revise the references accordingly?

Congratulations on a very interesting manuscript, and thank you for choosing WES to publish your excellent work.

Best regards,
Andrea Hahmann

Dear Dr. Andrea Hahmann,

Thanks very much for dealing with this manuscript! We have revised the final version according to your comments and revised the reference format following the guidance.

Regards,
Ye Liu